# Enabling Blockchain Services for IoE with Zk-Rollups

**DOI:** 10.3390/s22176493

**Published:** 2022-08-29

**Authors:** Thomas Lavaur, Jérôme Lacan, Caroline P. C. Chanel

**Affiliations:** 1ISAE-SUPAERO, University of Toulouse, 31400 Toulouse, France; 2University Toulouse III Paul Sabatier, 31062 Toulouse, France

**Keywords:** IoT, IoE, blockchain, rollup, Zero-Knowledge, zk-rollup, scalability

## Abstract

The Internet of Things includes all connected objects from small embedded systems with low computational power and storage capacities to efficient ones, as well as moving objects like drones and autonomous vehicles. The concept of Internet of Everything expands upon this idea by adding people, data and processing. The adoption of such systems is exploding and becoming ever more significant, bringing with it questions related to the security and the privacy of these objects. A natural solution to data integrity, confidentiality and single point of failure vulnerability is the use of blockchains. Blockchains can be used as an immutable data layer for storing information, avoiding single point of failure vulnerability via decentralization and providing strong security and cryptographic tools for IoE. However, the adoption of blockchain technology in such heterogeneous systems containing light devices presents several challenges and practical issues that need to be overcome. Indeed, most of the solutions proposed to adapt blockchains to devices with low resources confront difficulty in maintaining decentralization or security. The most interesting are probably the Layer 2 solutions, which build offchain systems strongly connected to the blockchain. Among these, zk-rollup is a promising new generation of Layer 2/off-chain schemes that can remove the last obstacles to blockchain adoption in IoT, or more generally, in IoE. By increasing the scalability and enabling rule customization while preserving the same security as the Layer 1 blockchain, zk-rollups overcome restrictions on the use of blockchains for IoE. Despite their promises illustrated by recent systems proposed by startups and private companies, very few scientific publications explaining or applying this barely-known technology have been published, especially for non-financial systems. In this context, the objective of our paper is to fill this gap for IoE systems in two steps. We first propose a synthetic review of recent proposals to improve scalability including onchain (consensus, blockchain organization, …) and offchain (sidechain, rollups) solutions and we demonstrate that zk-rollups are the most promising ones. In a second step, we focus on IoE by describing several interesting features (scalability, dynamicity, data management, …) that are illustrated with various general IoE use cases.

## 1. Introduction

Internet of Everything (IoE) can be defined as a generalization of the Internet of Things (IoT) which connects machines and people in one network [1]. It can also be viewed as the fusion of the Industrial Internet of Things (IIoT), which contains all the data related to industry, and the Internet of People (IoP), consisting of social networks and connections between people [2].

One of the main challenges related to IoE is handling the huge amount of data being generated. Their management is generally realized by centralized entities that can use scalable and efficient tools to provide better services to users. This has resulted in the emergence of giant structures in various fields (Internet, services, banks, …) and, thus, to a centralization of power in the hands of few entities.

Blockchain technology was initially introduced to bring decentralization to financial systems (e.g., Bitcoin [3]). The massive adoption of major blockchains since the launch of Bitcoin in 2008 has proven the value of this decentralized technology, especially in financial areas. Even if its initial and main application was in the financial domain, it has been applied in other areas, such as IoT [4,5,6], networks of vehicles [7] and marketplaces [8]. Blockchains can be used in these contexts to provide integrity, auditability, and accountability, to remove central points and to give the power back to consumers. The architectures and results achieved in those fields have suggested that blockchain technology could be a good candidate to mitigate single point of failure vulnerability while providing strong security and cryptographic tools [9].

However, the success of blockchain application in other fields involving small connected or moving objects, with embedded systems limited by low computational power and storage capacities, is less evident. One of the main reasons for this is weak scalability. Indeed, large blockchains succeed perfectly in providing security services in a decentralized way, but the quantities of processed data are much lower than the needs of the industries and improvements are often brought about at the expense of security or decentralization.

In this paper, we first detail the various recent proposals for the most promising scalability improvements being considered for small connected or moving objects (e.g., robots, drones, satellites, etc.) but also for the scalability needed to read, write and store larger amounts of data. The extraordinary recent craze around blockchain technology has lead to a large number of recent results and proposals made by startups and private companies, mainly in the financial domains. In this context, these concepts are rarely formalized, validated and published in scientific conferences or journals. This is why it is urgent, on the one hand, to synthesize these works in order to share them, and on another hand, to propose new applications not related to finance or economics. Thus, our first objective is to review these different proposals by explaining and comparing them. We do not intend to conduct a systematic review since the rest of the paper is dedicated to new discussion and new propositions regarding the structure and applications of rollups for IoE. The second objective is to propose solutions using these improvements to enable the practical use of blockchains for IoE. In particular, we advocate for solutions that are built around the zk-rollup structure, which appears to be able to solve the security/complexity dilemma. With the help of zk-rollups, it is now possible to use blockchain technologies that meet the performance criteria of the IoE without reducing security.

This paper is organized as follows: Section 2 introduces and defines the notions of IoT, IoE and blockchain and then presents the difficulties in connecting these domains. Section 3 presents the solutions to scale blockchains and remove obstacles to their adoption. Section 4 explains in detail the concept of zk-rollup, a concept that is not widely described in the scientific literature. Finally, in Section 5, we propose new structures and applications for zk-rollups connected to IoE that move away from the usual financial applications. Section 6 concludes the paper.

## 2. Blockchain and IoE: Promises and Constraints

Even though blockchains were initially built for financial applications [3], their properties can be of interest for countless IoE applications, such as IoT, marketplaces connecting suppliers and customers, social networks, etc. Proposals for non-financial applications have been made [10], but in practice, few are actually deployed. In this section, we recall the possible benefits but also the requirements related to general blockchains. Then, we consider some use cases and identify what the use of blockchains could promise and what constraints it could face.

### 2.1. Blockchains: Advantages and Requirements

A blockchain is a distributed and replicated ledger. It typically relies on a peer-to-peer network of nodes to communicate. To add a piece of information to such a ledger, often called a transaction due to the financial origin of blockchains, the sender signs it and sends it to the entire network. If a transaction is considered valid, it is added to the next batch of transactions within a block that is often size-limited. Then, a set of rules followed by the majority of the nodes allows for the election of the next node authorized to add the new block to the ledger. Everyone can check if this block is valid. This is the consensus of the blockchain. If two valid versions of the blockchain are received, usually the longest is kept, and the shortest is discarded. According to the context, it is possible to configure who can write and read on the blockchain, to define whether it is permissioned or permissionless [11].

A blockchain brings several advantages. It has the property of being publicly verifiable since all the rules are clear, common and fixed in advance. Even in the case where the right to write is reserved to a set of pre-established nodes, any observer can verify the validity of transactions. This provides transparency due to the ability of the nodes to verify the state of the ledger and ensure the validity of each transaction, which are state transitions and not necessarily linked to a financial statement. Moreover, the blockchain also allows for privacy through different cryptographic protocols such as Zero-Knowledge. Finally, the redundancy resulting from large-scale replication makes the system very resilient and almost immutable, guaranteeing data integrity [12].

Note that most of the applications in our daily lives are centralized by a few entities that manage and process our personal data to provide services. However, the user is no longer the direct manager nor the owner of his data. In the case of a bank, it provides financial services but also holds funds for its clients. In case of attack, corruption or reversal of this third party, our data (here banking data) can be corrupted, blocked or erased. The blockchain allows for the decentralization of applications, giving the power over their data back to the user [13]. The use of blockchains avoids attacks on single entities to recover all of their data, thus avoiding a single point of failure. Several examples perfectly illustrate this risk, like the leakage of health data of many governments, resold later by attackers [14].

Smart contracts are used by most blockchains. They are programs stored on the blockchain that can be called up via transactions; here, the term transaction refers to an execution order and, once again, not to financial transfer. The person chosen to create the next block including this transaction then runs the program, which can interact with the blockchain or return an output included in the block. To verify the proper execution of the program, the nodes that verify the block, re-execute the code and compare their results with those of the block. A smart contract is therefore a program written in the blockchain whose execution is included in the blockchain and verified by everyone. Ethereum, the second most-famous and most-used blockchain, was developed to respond to Bitcoin’s lack of programmability [15] and introduced smart contracts. To work, smart contracts on Ethereum are written in the Solidity language; once compiled, they can be executed with the Ethereum Virtual Machine [16] (EVM) environment as the java environment. Most blockchains enabling smart contracts rely on EVM. Smart contracts allow for more direct services by avoiding intermediaries, thus reducing costs, reducing the number of attack points for an adversary and allowing for more equitable power distribution.

However, the use of blockchains also brings with it negative aspects or problems to be solved. They are often less efficient than their centralized equivalents. The throughput, i.e., the number of transactions per second (TPS), of a centralized database is often very fast as the information is written without verification or after verification by the central entity at a single location and without communication with other participants. Blockchains are often much less efficient (from a dozen to a hundred TPS), and although permissioned blockchains are more efficient (on the order of a thousand TPS), these improvements are not enough to meet the needs of the IoE. Similarly, blockchains have a latency between the moment a piece of information is submitted, via a transaction, and the moment it is considered finished and included in the blockchain. This can vary from a few thousandths of a second for a centralized database to several seconds for a permissioned blockchain and reach latencies of upwards of several minutes [12]. For Bitcoin, if a piece of information is immediately included in a block, it takes an average of 6 more blocks (about an hour) for it to be considered definite. However, the number of writers on a permissioned blockchain is low and presupposes trust in this relatively small group. Performance improvements are often achieved at the expense of security or decentralization. This problem is often called the blockchain trilemma [17]. Table 1 inspired by [12,18] summarizes these issues.

### 2.2. Internet of Everything: Promises and Constraints

The term IoT refers to all connected physical objects with heterogeneous capacities. IoT incorporates both devices with significant capabilities, such as autonomous vehicles, and constrained systems in terms of CPU, memory and battery life. Such objects often have sensors that allow them to produce data, which are sent to remote devices or servers. Most of these communicate thanks to a centralized client-server model. This approach introduces some issues. For example, if the server that manages all the connected objects, linked to a geographical area or to an application, does not respond or becomes malicious, devices may exhibit unexpected behavior or disclose private information. We advocate that, in this context, a decentralized architecture would be more resilient to various failures or attacks [19]. Proposing a decentralized adaptive communication system while preserving a sufficient level of security is a main challenge that should be addressed.

Following the advent of IoT, the concept of Internet of Everything (IoE) has emerged. The idea comes from [20], which presents the “four pillars” of IoE, which are people, data, things and processes. The objective in interconnecting all these areas is to offer users intelligent applications that can analyze the data extracted from IoT. New intelligent solutions linking customers, providers and applications are based on this interconnection and allow improvements in many areas such as energy consumption, carbon dioxide emissions or better physical processing performance (see, e.g., [21] in the context of Intelligent Transportation System (ITS) applications). According to [22], three important points are expected. First is the scalability of the architectures required to allow objects and people to communicate quickly regardless of their environment. Secondly, it should enable the analysis of data to output intelligent actions or propositions. This requires a connection between local, distributed and cloud computing as well as a structure that allows for efficient, secure and easy-to-analyze storage. Finally, a wide variety of objects and applications must be supported regardless of their geographical location, environment, utility or technological requirements.

### 2.3. Difficulties in Connecting IoE to Blockchains

While a blockchain provides security through decentralization and immutability, it also brings up some issues. A natural idea in the application of blockchains for IoE (or IoT) would be to use a public and already well-established blockchain, such as Bitcoin or Ethereum. However, this is almost impossible for IoT. In order to maintain the security derived from the blockchain’s decentralization, each object must be able to verify that new blocks have been included correctly. A client (here, a connected object) would have to download the entire blockchain or at least the headers of the chain. Since the size of blockchains like Bitcoin and Ethereum is measured in hundreds of gigabytes, even when considering downloading just the headers, it is constraining for IoT and can be a real hindrance to its adoption. Additionally, clients participating in consensus would also be required to dispose of a certain level of computational power, in particular for Bitcoin or Ethereum, which use Proof of Work (PoW), that a common IoT device simply does not have. PoW relies on computing power: to be authorized to add a block of information to the blockchain, a node has to solve a difficult mathematical problem. The more computing power it has, the more likely it is to solve the problem quickly. The first node to succeed has then, very likely, invested economic resources in the form of energy and material expenses. It is therefore in its interest that its block be valid because if it is integrated into the blockchain, a reward is paid for solving the problem. If the block is invalid, the rest of the network verifying the proposal will reject it. It is also possible to use a “light” client, which allows one to download and store only the most recent data and the essential information of old blocks. Another solution is to directly connect remotely to an established blockchain node. However, this solution clearly reduces the decentralization and the security by requiring trust in an external entity.

Using a public blockchain as famous as Bitcoin or Ethereum could provide the greatest security, but could also pose obstacles for IoE. Information storage needs by connected objects and/or social network are abundant. However, the number of transactions, and thus the amount of data, that can be processed in one second on such a blockchain is very limited (7 to 8 for Bitcoin and between 8 and 15 for Ethereum). On top of that, the transaction fees for each message published on the blockchain are high. Consequently, it is impossible to store all communications directly on the blockchain. In a large group of connected objects, it is rare that each device would need to communicate with every other device. Often, groups are formed according to certain characteristics such as their position, their application or their interdependence. Creating groups relying on a dedicated blockchain could allow the system to be scaled up when the amount of communication and then transactions increases. This is especially evident for drones, smartphones and vehicles that do not maintain a connection with all others due to their environment. For memory storage scalability, not all transactions storing pieces of information need to be kept forever and immutably; this would prevent scaling.

All these issues are challenging for blockchains in general. Improving or resolving these problems is the main topic of several publications. However, it can be observed (see [17]) that the scalability of the storage size of blockchains is not well-studied and requires more research to enhance blockchain applications for light devices.

In response to these barriers, a second idea would be to use a private blockchain for each group. A private and local blockchain reserved for specific uses, or for a specific application, is vulnerable if someone with a higher consensus power gets into the network and controls the consensus. This is the case for IoT if the consensus used is PoW, for example: Connected objects have little computing power, while an adversary can infiltrate the network with a powerful computer or dedicated hardware. A public blockchain is too heavy for IoT which does not have enough consensus power to really participate in the network, nor enough storage and computing space to verify the integrity of the blockchain. Moreover, a transaction can be expensive due to the competition between nodes to include their data in the next block. To guarantee decentralization, any device must have access to the validator and block producer roles. However, scattering the objects over different, separate blockchains also disperses security by diluting the number of active agents checking the state of the blockchain and then reduces decentralization. At the same time, it makes interoperability between objects more complicated if they have to go through different blockchains or to change groups regularly.

Finally, an ideal solution would be to find a compromise between the strong decentralization, security and high adoption of a public blockchain and the light, fast, cheap transactions and small group division that a private blockchain allows.

## 3. Scalability Solutions

Several levels of action can be used to improve scalability [17]. Most of the research focuses on increasing the number of TPS, and for this, it is possible to modify the protocols of the network itself, the consensus, the internal structure of the blockchain or add an external structure. Most of these scaling solutions can be classified into four different categories: network solutions, consensus solutions, on-chain solutions and Layer 2 solutions.

### 3.1. Network Solutions

Optimizing network communication for faster information propagation is essential for scalable and efficient exchanges, and as a consequence, it may increase the number of TPS. Most of this research is based on data compression and the redesign of the network structure [17]. For IoT, bandwidth can be physically limited, so it is important to reduce the size of the data propagated within the network to allow objects to continue participating despite their limitations.

### 3.2. Consensus Solutions

Blockchain’s consensus can be seen as the way that involved participant nodes agree about the validity of data and how the data will be stored in the blockchain. There are many consensus protocols, but the two most well-known for public blockchains are Proof of Work (PoW) and Proof of Stake (PoS), which replaces the election of the block submitter based on a lottery system where the more tokens (the currency of the blockchain) one holds, the more chances one has to be chosen. Once again, PoS allows the network to incite the creation of valid blocks by remunerating their creators and removing a quantity of their tokens if the blocks are invalid. These two protocols are very different, but multiple improvements have been proposed to enhance them [23]. The idea is that the consensus can be adapted in order to make the creation of new blocks accessible to everyone, thus increasing the throughput (TPS).

For IoT, it is obvious that PoW has to be avoided as its security is based on computing power, which is limited in most connected objects. PoS can be used by IoE but leads to worse decentralization. Most IoT that include blockchains use the Proof of Authority (PoA) consensus model based on the identity and reputation of participants. However, this form of consensus is even more centralized and less secure, albeit faster in terms of TPS than PoS. The question of choosing the correct consensus is a complex one and must be based on the needs of each entity group, i.e., according to the number of participants, their overall and individual capabilities and their application.

### 3.3. On-Chain Solutions

Restructuring information on the blockchain and redefining the blockchain itself can be a solution to increase the number of TPS or to decrease the size of blocks. A natural idea to increase throughput is to directly increase block size. However, this has the direct consequence of increasing the size of the blockchain, impacting the participation of light devices and nodes with limited storage space. Another approach is to redefine what a transaction is in order to reduce its size, enabling a block to contain more transactions. This idea is particularly interesting for fiduciary transactions or for the storage of short messages. The majority of the weight that a transaction takes on (in number of bytes) comes from the user’s signature. It is possible to separate signatures from transactions and compress them, as suggested by [24].

A complete change in the internal structure of the blockchain may also be a way to increase the number of TPS. A first idea, introduced in 2015 with the DagCoin cryptocurrency [25], is to replace the linear, chronological structure of block addition (all in a row) with a direct acyclic graph (DAG). In this new architecture, when a block is added, it does not refer to the previous block as in a classic blockchain, but to at least two previous blocks. The throughput is increased since it is possible to create several blocks in parallel. The main drawback of this structure is that it is necessary to reexamine the security model to avoid double-spending (i.e., double-writing). As the DAG adds blocks faster, the system also grows faster, thus increasing the required storage space and the risk of centralization once again since the required storage space for nodes would increase. The DAG IOTA [26] was introduced in 2016 specifically to address the problem of TPS and is often proposed for IoT. However, IOTA was extremely ambitious and introduced many new technologies that have not been fully validated, especially for the consensus and security aspects [27].

Another idea inspired by distributed databases is the segmentation of the blockchain into groups. This segmentation, first proposed in 2016, is called sharding [28]. Each group, called a shard, acts as a blockchain by performing transactions and hosting data. The consensus is executed in parallel for all shards and they can communicate with each other. This solution is very interesting because it allows the system to distribute the storage space and the execution of the transactions, i.e., state transitions. However, the consensus must be carefully studied to avoid diluting security between shards. A new type of attack appears with this architecture because the whole network becomes vulnerable if only one shard is vulnerable. The attack aiming at compromising a shard is called the 1% attack [29].

Finally, several ideas have not yet been implemented but are under consideration for addition to the Ethereum blockchain. This is the case for EIP 4844 [30]’s blob-carrying transaction, which is a temporary transaction whose data can be removed from the blockchain after a certain delay. This can be used by IoT with specific applications such as mapping to keep track of recent data and remove old useless transactions.

### 3.4. Layer 2 Solutions

#### 3.4.1. State and Payment Channels

A state channel or payment channel is the opening of a communication channel outside of the blockchain. More precisely, a state channel in general is a channel that allows to update a state of information, and a payment channel is used for fiduciary transactions. Payment channels are thus state channels dedicated to financial transactions. The transactions are carried out and only the summary allowing the move from an original state to a final state is posted on the blockchain. This allows us to overcome the limits of the blockchain while taking advantage of its immutability property. However, the consensus cannot guarantee the state changes as long as the state has not been updated on the blockchain, and a third party can act maliciously on this external channel until the final state is published. The lightning network [31] and Raiden network [32] are the most well-known payment channels, built, respectively, on Bitcoin and Ethereum.

#### 3.4.2. Sidechains

All previous proposals show that it is difficult to manage very large amounts of data and transactions on a single blockchain. This is, firstly, because public and permissionless blockchains do not have enough TPS, and because permissioned blockchain are more centralized. Previous proposals mitigate these issues but lead to a reduction of security and a risk of centralization. When the volume of data stored on the blockchain increases, the risk of centralization increases as well, and it becomes more and more difficult to keep a copy of the blockchain on a light or classical system. The intuition to use a different blockchain for each individual application, need or group could then arise naturally, here.

To guarantee security, a so-called two-way bridge system must be set up to perform transactions from one blockchain to another. This is called either a cross-chain protocol when the two blockchains already exist, or a mainchain and a sidechain if a second blockchain (the sidechain) is created in addition to an existing one (the mainchain). A sidechain can have its own consensus system, obey its own rules and rely only on its own security.

First introduced in 2014, two-pay peg [33] is a way to transfer money or information from one blockchain to another. The idea is to use a trusted third party operating on both blockchains. By sending a transaction on the main chain via a lock-box address to the trusted third-party, as soon as it is confirmed, the funds can be released or the information propagated on the sidechain and vice-versa. It is also possible to distribute the trust of this central authority among several participants, who form a committee of *n* persons. To carry a transaction, it would be necessary for *m* among the *n* members to accept it. This is the federated two-way peg as used in RootStock [34], a Bitcoin sidechain that enables smart contracts on the main chain (Bitcoin, in this case). Finally, it is also possible to automate this exchange with smart contracts and to completely decentralize them by using a dispute system (see Figure 1). The decentralization of the two-way peg protocol involves a proof of deposit and/or a dispute phase during which each user can challenge the veracity of the initial transaction. If a transaction is fraudulent, it is canceled, and a reward is given to the user who reported it. This phase often lasts a long time. To prove the proper execution of a transaction, a Merkle proof [35], which ensures that the transaction belongs to the set of transactions executed on the sidechain, can be used. If the user does not prove the execution, they are punished. After a fixed time without being contested, the transaction is relayed to the destination blockchain. This saves TPS and storage space by using only the blockchain that meets the application’s needs. But this gain comes at the cost of decentralization because most participants will choose which blockchain(s) to participate on. Security may consequently be reduced.

Note that, in these schemes, if the sidechain is replaced with a central authority or a different secure structure, the payment or state channel structure is recovered.

#### 3.4.3. Plasma Chains

Introduced in 2016, Plasma [36] reinvents the organization of sidechains. The Plasma structure is based on the use of smart contracts and Merkle trees, allowing the creation of several sidechains. It is then possible to create a sidechain of an existing sidechain. The nomenclature adopted for this is: child chain, parent chain and root chain (the main blockchain). Each chain is designed to operate individually and independently, accommodating different needs. The creation of several child and parent chains produces a tree structure. A decentralized two-way peg model is used to transmit data from one chain to another, and a state commit is regularly sent to the parent chain, consisting of the block header. More precisely, it is a decentralized model, based on fraud proofs, which allows users to report instances of malicious activity and protect their funds (i.e., stored information). Evidence of fraud allows a Plasma child chain to make a complaint to its parent chain or to the root chain and recover its funds.

Plasma therefore brings important improvements related to scalability, such as increasing TPS or reducing the memory space needed to read and verify a blockchain. However, this model also has drawbacks. Such a model requires setting the characteristics of each new child chain as with the creation of an independent blockchain. The system based on fraud proofs allows any user to secure their funds and information because they will always be able to transfer them to the parent chain even if the child chain consensus has broken. However, this process is lengthy; a user can wait between 1 and 2 weeks before the dispute phase is over. Moreover, the multiplication of chains can raise security issues with respect to consensus. Taking the Proof of Work consensus as an example, the multiplication of different blockchains, all relying on the computing power of the network, can lead to a decrease in security because of the reduction of the consensus power allocated to a specific chain. Indeed, the nodes will have to choose which consensus to participate in and this implies a distribution of computing power amongst the different blockchains. This is related to the centralization that the multiplication of child chains brings.

#### 3.4.4. Rollups

Rollups are a Layer 2-type solution that extends the ideas of sidechains. This technology is evolving very quickly, and its deployment is very much linked to business. Unfortunately, it is difficult to cite peer-reviewed publications dealing with this subject because the main innovations in this promising domain come from the private sector. Companies making advances in this field are aiming at being the first to propose this kind of service, and they must not have a strong interest in or simply do not have the time to go through the process of publishing their research in scientific conferences or journals.

A rollup relies on the security of the blockchain it is built on. All transactions are stored on the blockchain but are executed off-chain by a central entity. Rollups increase throughput by reducing the size of each transaction and by removing the need for execution on Layer 1: Only the verification is done on the blockchain. To create a rollup, a smart contract first builds two associated Merkle trees: One to store the accounts, as shown in Figure 2, and one for the funds. The trees are not directly stored on the smart contract but can be rebuilt from the transaction history. The first one makes it possible to keep a record of the address of the accounts it manages in memory, and the second the record of the balances of the accounts on the corresponding leaves. This construction reduces the size of the transactions performed by the rollup because the destination address will be referenced by its index on the account tree and enable the smart contract to only store the roots of the Merkle trees. Figure 3 shows a transaction on the Ethereum blockchain incorporating a rollup transaction. All transactions of the rollup are always accessible from Layer 1 (i.e., the main chain) as they are all stored in a specific field of the main chain’s transactions, called the calldata. Calldata is a non-verified part of an Ethereum transaction and is not size-limited. Calldata can store any type of data, and is cheaper (fewer resources are required to execute this part, rendering it less costly) since it only requires storage and not verification. A Merkle proof can also be sent to the smart contract in order to prove that an account has funds.

As explained, the biggest parts of a transaction on the Ethereum blockchain are the signature and the destination address. The use of the Merkle tree allows for a reduction in this address from 20 bytes to only 3 by using indexes (see Figure 3). The signature is also no longer necessary because an anti-fraud system is in place. In fact, the signature is replaced by the index of the emitter’s address that was habitually recovered from the signature.

Two types of rollups exist to avoid fraudulent transactions while deleting signatures. On the one hand there are optimistic rollups [37], which assume that all transactions are honest and in case of suspicions, a dispute system can be used. The node responsible for updating the Merkle trees must then justify itself by providing the signature linked to the transaction. If the suspicion is founded because no justification is brought, the user who reported the transaction is rewarded; otherwise, they are sanctioned. On the other hand, zk-rollups prove a batch of transactions with one Zero-Knowledge verifiable computation proof [38]. The proof is included directly in the calldata after the transactions. Both of these two different systems have their advantages and disadvantages. Optimistic rollups are low-cost because no extra work is required. However, the dispute system can lengthen the time to withdraw funds from the rollup upwards of several days. Zk-rollups do not have this problem since all transactions are proven directly at the time of the Merkle tree update. However, they require an additional workload through the computation of the proof, which can be substantial. The Zero-Knowledge proofs used are also difficult to adapt to smart contract execution, while the fraud system would simply need to be adapted slightly for optimistic rollups. In Section 4, a more detailed explanation of how a zk-rollup works is proposed. Optimistic rollups work similarly but use fraud proofs instead of Zero-Knowledge proofs. The whole process is illustrated in Figure 4.

## 4. Zk-Rollups

### 4.1. Verifiable Computation

Verifiable computation, also known as proof of computational integrity, is a cryptographic protocol allowing a user or a client to prove a calculation, or more generally, a program’s execution by computing a proof. The proof is then provided to a verifier who can ensure that the program has been executed correctly.

To do this, almost all verifiable computation schemes follow the same protocol. First, they transform the function that must be proven into bounded-degree polynomials [39]. This step transforms the discrete steps of a program where an error is local and hard to detect into polynomials that will undergo many changes if a step is not correct or respected. The error is then much easier to detect. However, simply checking these polynomials by giving them to a verifier is not easier than redoing the computation of the program itself. The hardest part is then cryptographically proving that the polynomials are indeed of low degree without revealing them. If the polynomials were not of low degree, the prover would have been able to choose the polynomials after the execution of the program and create a fake proof, breaking the protocol. The evaluation of polynomials on random points chosen by the verifier or a random oracle (often a hash-based pseudo-random function emulating the unpredictable behavior of the verifier) and proving that the correct polynomials have been used without revealing them is the difficult concept that most protocols differ on.

Most verifiable computation schemes can be classified into one of two categories: SNARKs and STARKs. These categories are presented below.

#### 4.1.1. SNARK

The Succinct Non-interactive ARgument of Knowledge (SNARK) allows providing a verifier with a succinct proof, i.e., a proof that can be verified in a logarithmic time. This type of scheme has known many improvements since the first practical protocols [40]. In particular, Groth [41] allows providing proofs of fixed size (less than 200 bytes) which are verified in constant time regardless of the program to be proved. The main disadvantage of this scheme is that it relies on a trusted setup that must be done for each different program, resulting in a Common Reference String (CRS), also called a verifier and prover key. The verifier key is sometimes different from the prover key because precomputations can be done for the verifier. The prover key is the same as the CRS.

Even though SNARKs are subject to a trusted setup, it is possible to separate the trusted setup into two parts: one dependent on the program to be checked but without trusted third parties and one independent of that, on which security is based. This independent part can be built with several participants, distributing the security amongst them, and can be used for any program up to a certain amount of operations (thanks to the powers of tau ceremony [42,43]). The result of this multiparty protocol is the same as in a classical trusted setup, but the trust is no longer based on a single participant. The idea is that security is ensured if at least one of the participants has been honest during the ceremony. Reusing completed ceremonies that have included many well-known personalities helps to build confidence in this model.

#### 4.1.2. STARK

The Scalable Transparent ARgument of Knowledge (STARK) [44] differs from a SNARK by the absence of trusted setup. Scalable also means that in addition to being logarithmic in verification time, like SNARKs, STARKs must also be at most linear in proof time. Moreover, STARKs rely only on hash functions and Reed–Solomon codes’ [45] properties, which makes them post-quantum (post-quantum cryptography aims to guarantee the security of information against an attacker equipped with a quantum computer). However, STARKs are recent proposals and more research is still needed to be competitive against SNARKs in certain areas, as in blockchains more precisely, due to their proof length.

#### 4.1.3. Discussion and Comparison

A step during the verifiable computation protocol can easily be added to SNARK and STARK to keep a part of their public inputs secret. Proving the knowledge of a piece of data without revealing it is called Zero-Knowledge proof in cryptography. Once Zero-Knowledge is added, these protocols are refereed as zk-SNARKs and zk-STARKs.

However, proving a program requires additional computational efforts that are not negligible. The arithmetization phase (when programs are transformed into polynomials) pushes programmers to rethink the program that had been optimized for a given application. Let’s take the example of hash functions. Currently, they have been designed to be executed as quickly as possible by a CPU and are therefore oriented on Boolean operations. However, arithmetic makes the transformation of these Boolean operations difficult and expensive, making the calculation longer and more expensive. It is therefore necessary to re-imagine programs by orienting them on arithmetic operations that are easy to prove and to optimize the combination of computation and proving time. Current research also focuses on the development of new “SNARK-friendly” cryptographic tools that would reduce the cost of Zero-Knowledge proofs, such as hash functions [46,47,48] or new elliptic curves [49].

For blockchains, SNARKs are better suited for verification because the proofs are significantly shorter. STARKs have the advantage of being asymptotically faster to produce than SNARKs, but even though SNARKs are not post-quantum, most current blockchains are not either. Additionally, SNARKs have a broad advantage for blockchains: They allow calculations that come from outside the blockchain to be verified quickly on the blockchain. It is a secure way to import information without relying on a trusted third party like classical oracles.

Moreover, proofs of computational integrity can be a real advantage for IoE applications, because they allow for the delegation of computations that are too heavy for light systems to servers or other more powerful machines. Small devices can then verify the result by checking the proofs of computational integrity related to their computation. Research is particularly active in the effort to improve the verifiable computation and its various techniques such as [50] that reduces the overhead of proving the computation. Other improvements address specific subjects such as smart-contract-compatible zk-rollups [51] and recursive verifiable computation proofs [52], which are proofs of the verification of a proof. In the following, the principle of the zk-rollup concept is introduced.

### 4.2. Zk-Rollups: Principle

The zk-rollups concept is presented in this section via the example of a transaction cycle using a zk-rollup on the Ethereum blockchain.

First, a smart contract is deployed on the blockchain. Its role will be to maintain the state of the accounts via the roots of two Merkle trees and to verify state transitions. It may contain the verifier key able to verify one of the Zero-Knowledge proofs described in Section 4.1 if needed (such as SNARKs, for instance).

Now, suppose that Alice wants to send 5 ETH to Bob using a zk-rollup and that Bob already has an account on Layer 2. Alice first sends the ETH that she wants to deposit on the zk-rollup and that can be used on Layer 2 at the address of the smart contract, ➀ in Figure 4. The deposit is stored in a queue to be inserted by a prover. When the accounts are updated by the prover on the smart contract ➁, an index is assigned to Alice’s public key in the Merkle tree of accounts, and the amount she paid is added to the same index in the balance tree.

A prover is then chosen to validate the next zk-rollup transactions. Either they are designated by the smart contract, elected by an auction system or by any incentive system like Proof of Stake (PoS) or Delegated Proof of Stake (DPoS). Once designated, if the prover does not respond within a certain time-limit, they are sanctioned, and a new one is designated to replace them. A malicious prover can only freeze the account status temporarily, and in case of misbehavior, they are penalized, and another one takes their place.

Once the prover is chosen, the prover receives the transactions from a transaction pool as with a classic blockchain. Thus, Alice just sends her transaction to the prover via the peer-to-peer network, saying she wants to send 5 ETH to Bob’s rollup account by including his index ➂. The prover receives the fees included in all transactions they prove. After a fixed amount of time or when they reach the maximum number of transactions, they use the Zero-Knowledge arguments to prove the update of the Merkle trees’ root. They prove that the transition from the old state of the accounts to the new one is correct, i.e., the signatures of all the transactions are valid—note that the proofs make signatures useless on the blockchain—and are included in the transition from the previous state to the new one. Finally, the prover sends the updated accounts to the smart contract via a single transaction. To do this, it includes all the zk-rollup transactions in the format presented in Figure 3, the roots of the updated Merkle trees and the proof of its calculations in the calldata ➃.

The trees can always be reconstructed outside the blockchain by looking at the transaction history. When the smart contract receives the transaction, it verifies that the proof provided is correct by using the previous transactions and account states as the public input. If the proof is correct, it can update the trees’ root. At this moment, Alice can detect the change and see that her transaction was included on the blockchain. Finally, Bob sends a new transaction on Layer 1 to ask the smart contract to withdraw his funds ➄.

Some rollups split the responsibility for publishing data to update the accounts status into two parts. The batch of transactions and the root of the two Merkle trees are not published by the same actor. This scheme has two main advantages. If the roots of the Merkle trees and their proof are invalid, it is not necessary to re-publish the transactions but only to evaluate and re-submit the proof. Several actors can post batches of transactions in order to improve censorship resistance while avoiding an issue where certain groups of transactions would be invalid because others have been published before them.

Currently, as of 25 July 2022, the cost of a classical transaction on the Ethereum blockchain is $1.30 to send ETH, while on zkSync [53], a zk-rollup based on SNARKs, this cost drops to $0.05. For a swap token transaction, the cost rises to $6.49 while maintaining a reasonable price of $0.13 on zkSync.

### 4.3. Optimistic Rollups vs. Zk-Rollups

Zk-rollups were the first to be introduced in 2018 by Bary WhiteHat [38]. However, these are hard to expand for the purpose of proving smart contracts using SNARKs or STARKs. It was to answer this difficulty that in 2019, John Adler proposed optimistic rollups [37] (see Section 4) that use the scalability improvements of zk-rollups and combine them with the fraud proofs of Plasma instead of Zero-Knowledge proofs. They allow for the direct execution of smart contracts without modification outside of Layer 1. However, the fraud proofs also bring their own set of disadvantages. It will thus be necessary to wait between 1 and 2 weeks on average to be able to pass the dispute phase and to recover funds through the smart contract of the blockchain.

The technology is evolving rapidly, and zk-rollups should be able to support smart contracts in the near future. Currently, Solidity and Cairo [54] smart contracts are the only ones supported by zk-rollups, with few unsported Solidity opcodes [55]. Only small modifications will therefore be necessary in order to compile native Ethereum smart contracts on zk-rollups. Indeed, complete Turing languages already exist, such as Cairo [54] developed by Starkware. Several approaches aim at implementing a proof system for smart contracts, such as that of Polygon Hermez [56]. A public testnet is available for zkSync 2.0 [55], an upgrade of zkSync which supports smart contracts. Several organizations such as Polygon and zkSync have already announced that they will deploy zk-rollups supporting smart contracts on the Ethereum mainnet in the near future. As of July 25, 2022, zkSync has announced that its upgrade is set to take place in approximately 100 days.

It is also important to mention the Mina [57] and Celo [58] blockchains. Mina is based on recursive proofs to obtain a blockchain of fixed size by proving the transition state from the previous block, maintaining a size of 22 kB. While zk-rollups prove transactions, Mina also proves the consensus and resolves the problem of choosing between two forks of a blockchain without knowing which one is the longest. Celo enables light nodes to verify the entire blockchain briefly by regularly making checkpoints that prove all preceding history. Celo aims to bring light clients to mobile phones and enable verification of the blockchain by light devices.

To conclude, optimistic rollups are easier to develop due to their low technical complexity compared to zk-rollups. However, even if zk-rollups require more computing resources, they also bring smart contracts to Layer 2. Zk-rollups are gaining in popularity due to their better performance, improving the transaction rate of optimistic rollups by up to 10 times. Additionally, they also leverage gas cost per transaction more than optimistic rollups and enable fast withdrawal.

### 4.4. Data Management

A zk-rollup stores all transactions on the main blockchain, which allows anyone to be able to reconstruct the Merkle tree and thus build a Merkle proof that an account has the money it claims at any time. However, despite the many scalability improvements brought by zk-rollups, the transaction fees or the number of TPS remain too expensive or too low, respectively, for some applications, such as, for example, games that need a huge amount of TPS and data storage.

As with two-way peg, it is then possible to use a less decentralized system than the blockchain to manage these data. Indeed, one can opt to delegate the management of data storage to a trusted third party, a trusted group or even an external blockchain. In exchange for reduced security, since access to the data guarantees access to its funds, it is then possible to gain in scalability. This method of storing data externally is called validium or is sometimes included in the term zk-rollup. The first name of this structure was Plasma with SNARKs to refer to Plasma but replacing fraud proofs with Zero-Knowledge proofs. In the rest of this paper, the term zk-rollup will be used for the classical scheme, and validium will be used when the data are stored outside of the blockchain. In the context of IoE in particular, the information that is important, with respect to the application, must be uploaded to the blockchain, while other information must not necessarily need to be stored on the blockchain.

It is possible to choose which transactions are to be registered on the blockchain, which are to be stored on a replicated database within a committee or by a single central authority. Note that validium stores every transaction in an external database(s), but when it is possible to decide for each transaction whether it is stored in the database or on the blockchain, it is called volition. In the case of an attack on the validium database, the attacker can only take the data hostage and freeze the validium. With a consensus, it is then possible to start over from an older state of which a user has the copy with new managers [59].

In the case of separation of each application in different rollups, the main difficulty lies in the possible dispersion of data and funds. In a zk-rollup or a validium, double spending is avoided because the money is first locked on a smart contract responsible for the rollup state, then released on Layer 1 once the transactions are completed on Layer 2. The transactions cannot spend the same resource on both levels at the same time. In the context of multiple rollups, it would be disadvantageous to disperse user funds according to each application or geographical area. It is then necessary to find a different architecture or rethink the rollup in order to limit this dispersion while avoiding the double spending attack.

## 5. Enabling the Decentralization of Ioe Applications Using Zk-Rollups

This paper advocates for the use of zk-rollup approaches for IoE applications, relying on mathematical proofs rather than relying on the good behavior of users, as optimistic rollups do. We believe that zk-rollups can be the solution that enables the use of blockchains for IoE thanks to their scalability improvement and their strong security equal to that of their Layer 1 blockchain. Moreover, zk-rollups are more efficient and the Ethereum blockchain plans to develop on a zk-rollup-centric roadmap. However, the paper already discussed that optimistic rollups are easily compatible with smart contracts. Therefore, one could propose modifications to be applied to zk-rollups, expecting that these could also be used in enhanced-optimistic rollups. Thus, in this section, we present 3 types of contributions. The next paragraph first describes the classical application of rollups and lists all the benefits of this solution for IoE applications. Then, we propose original adaptations of rollups to fit some IoE constraints and thus to enlarge the set of IoE that can use this technology. Finally, we describe several general IoE use cases that could be deployed based on adapted or classical zk-rollups.

### 5.1. Benefits of Using Rollups for Some IoE Applications

It can be noted that some IoE applications can directly benefit from the rollup architecture. Rollups allow the construction of a closed or dedicated network, at the same time reducing the transaction costs of a classical blockchain. They also enable easily deployment of a secure network based on the already well-established security of a blockchain with many users. Current developments will soon make it possible to use smart contracts directly on the zk-rollup and thus to use it in exactly the same way as a dedicated blockchain. The benefits can thus be multiple since it is possible to create a rollup specific to a group of users or to an application, in the manner of a blockchain but without redefining the security layer. At the same time, we can rely on an existing user panel and an already established and distributed currency and focus only on the needs of the created application or group.

The use of rollups by low-capacity objects reduces the necessity for participating in consensus since the exchanges are no longer secured by the consensus of the rollup but by the underlying blockchain. In the case of zk-rollups, security is ensured by the blockchain and the Zero-Knowledge proofs. The verification of these proofs has a very low complexity. For example, in [41], the verification complexity is constant (only a few operations on elliptic curves) regardless of the program. It also enables a reduction in the memory space needed to ensure that a transaction has been included in the blockchain. Indeed, a connected object can ask for a proof attesting that the hash chain is valid through to a certain recent block of the blockchain, as Celo [58] does, and check only the end. It works like a light node for which the chain does not grow because, as soon as it exceeds a size *n*, a Zero-Knowledge proof can compress *n* blocks. Zk-rollups enable this succinct database, since users only need to store the actual state of the rollup and verify transition proofs.

At the same time, zkSNARKs and zkSTARKs can enable the realization of computations that are too costly by delegating them to a third party thanks to rollups supporting smart contracts called zk-EVM zk-rollups. The proof will guarantee that the calculation has been done correctly and can be verified either by the blockchain or by the underlying smart contract-compatible zk-rollup.

### 5.2. Adaptations of Rollups to IoE Constraints

In the rest of this section, we propose revisiting the use of rollups beyond the scope of financial applications. As far as we know, rollups are used exclusively for financial applications and in this section, we will present new ideas allowing for more general use from which the IoE can benefit. We propose studying zk-rollups instead of optimistic rollups since the resolution of fraud proofs can take up to a week which is not at all suitable for IoE.

#### 5.2.1. Including General States in Rollup Merkle Trees

The applications of rollups currently deployed on blockchains are, to our knowledge, mainly financial. However, a rollup can be seen more globally as a tool allowing users to reconstitute states from the compressed transitions stored on the blockchain. These states do not necessarily have to describe financial statements. In fact, they can represent any general state linked to an account representing a user or a piece of data. It is possible to have several pieces of information related to an account stored in a single rollup as long as the rules for managing and updating these data are known and provable with verifiable computation. In the case of financial transactions, the proof ensures that the signature is from the sender, that the necessary amount is available in their account and that the receiver exists. However, for general use, the rules to validate a transaction can differ and need to be adapted.

In the context of IoE applications, the verification can include behavior consistent with a rule fixed in advance. A rollup could link an account representing an entity (an IoT device, a package or a service, for example) to a piece of information (its position, its delivery state, or its value). Several pieces of information can be linked to the same account and not only the funds it has. Since transitions must be proven by a system of fraud proofs or Zero-Knowledge proofs, the possible transitions must be fixed at the rollup deployment in the smart contract.

To store several pieces of information related to the same account, it is possible to create a new Merkle tree for each type of data to be inserted at the position corresponding to the position of the owner’s account leaf in the address tree. This allows to add information related to addresses independently, and their updates can be done in parallel. It is also possible to optimize the memory space taken up by this information and keep a rollup based on two trees: the first one for the addresses and the second one to store all the other data.

For example, one can imagine an IoT system where each leaf of the account state tree is replaced by a more global state tree. This one can, for example, include the funds as with a classical rollup, but also the position in three coordinates, the state in use or pending and its reputation. In IoE, for a social network, each account can have an associated reputation, number of likes and followers, etc.

#### 5.2.2. Tree of Rollups

If a user wants to exchange on several rollups or at the same time on the rollup and the blockchain, they have to split on each. It is obvious that if each person had to have a bank account with each store where they make purchases, it would be a huge hindrance to the use of such a system. Some protocols try to solve this problem by allowing the transfer of funds from one rollup to another without having to go through the blockchain, like Hop bridge [60]. However, bridges often rely on central authorities, diminishing the security provided by zk-rollups and blockchains. New security flaws may also appear in the smart contracts enabling them. This is why we propose to consider a tree structure, as in Plasma [36].

To extend the rollup structure to multi-layer use, we propose not only using rollups as a Layer 2 but also to extend their use to a Layer 3 or higher. To do this, it is possible to modify the structure of rollups. It is noticeable, given the important diversity of the networks relating to IoE, that the structure of rollups will have to be thought of differently for each application, in the same way as smart contracts. For that, we take advantage of zk-rollups compatible with smart contract (i.e., Ethereum’s zk-EVM) technologies to build rollups on other rollups. Indeed, a multi-layer zk-rollup must necessarily be able to prove the execution of the smart contracts, which is typically the feature of zk-EVM. Note that some applications need to have a management of large (e.g., geographical) areas while delegating part of the operations to its sub-regions or for a social network to manage sub-categories. These ideas are summarized in Figure 5.

It is worth saying that data management must be done in a different way. It is difficult to upload the data of all the rollups on the blockchain if their numbers increase, especially if some data are less sensitive and do not need a high level of security. To cope with this issue, we propose the use of a volition model where, when a user sends (or receives) a transaction for a change of state, they choose if they want to upload the information to the blockchain or not. Transaction fees can then be proportional to the number of layers that the transaction must go through.

#### 5.2.3. Dynamic Groups and Data Management

The rules and information incorporated in the rollup could take into account the creation of independent groups and their merging in the context of mobile connected objects. All of this makes it possible to offer a network that is independent of the blockchain and still relies on its security. In several types of applications, the creation of an ephemeral network is necessary. Moreover, in this context, it is often not necessary to upload the data on the blockchain. This kind of data management could be very interesting in the development of smart cities. To handle this, this work suggests a structure that can be used for rollups. These should be malleable thanks to the ease of the creation of separate groups within the rollup, the grouping of users in a single group, or the temporary creation of a subgroup.

In addition to that, a new management of access rights needs to be incorporated into the rollup architecture. As with blockchains, it is possible to choose the rights that a user of the rollup has. In this sense, this work will talk about permissioned or permissionless rollups (as with blockchains). The idea is to use permissionless rollups if everyone can read the data state and write, and permissioned rollups if everyone can read but only authorized users can submit changes. Considering the properties of validium, an additional restriction to link the properties of private and closed blockchains to the security and exchange of a public blockchain could be considered. By creating a permissioned validium, one can also restrict the access to read states to certain members with authorization. The validium thus acts as a private distributed database whose changes are proven and verified by the blockchain. The advantage is that the verification induces a higher security and that it is possible to upload data to the blockchain easily and securely. To make a piece of validium data public on the blockchain, it is sufficient to prove that it is part of the validium without revealing anything about the other pieces of information. If the state corresponds to a Merkle tree, it is therefore enough to reveal the element and to give a Merkle proof proving validium membership.

### 5.3. Use Cases

The use cases presented in the following are quite large and will not go into specific details. The main objective of this section is to provide an overview of all the interesting properties of rollups by offering a variety of proposed use cases. We believe that each specific case could be studied in more detail and developed according to the constraints of the target application. We will use zk-rollups built on Ethereum as an example, but these can be built on any Layer 1 supporting smart contracts or the verification of Zero-Knowledge proofs or fraud proofs.

#### 5.3.1. Deployment of a Global Service

A global, i.e., large-scale service, such as a taxi or delivery service, needs a large amount of communications and transactions. Moreover, the use of a blockchain can provide many benefits, whether it is security by avoiding the centralization of data by a private multinational company or the decentralization of the currency by relying on a crypto-currency. It can also bring transparency as the algorithms used are known to all and there is no discrimination possible as users and suppliers are all treated the same way. At the same time, anonymity can be reinforced by using mixers that allow users to change their identity on the blockchain [61]. In the following, we propose two ways to implement this kind of application with zk-rollups.

The first one is based on the structure rollup trees introduced in Section 5.2.2. It can be observed that such a large amount of information, whether or not it is only used for issuing offers and requests, cannot be included in the blockchain. Assuming that a rollup, a validium or a volition is deployed to meet this demand, even if it is specialized for such an application, the amount of state changes may be large. We suggest using the tree structure that would allow the creation of a rollup/validium/volition with several layers. For a large scale application, it is possible to create a state of the accounts which is an intermediary to other rollups to redistribute the accounts’ funds over different geographical areas. One can easily imagine a tree structure where the root is the application, which then separates the accounts by continents, then by countries, and, finally, by regions. This structure must be implemented with zk-EVM-compatible rollups.

In the second way to implement global services with rollups, we consider the possibility of putting it all together in a single zk-rollup. The account status would then include, in addition to the information needed by the applications, an index listing the sector to which the user belongs. Each account could then only exchange funds with accounts on the same index but could exchange information that does not require double-spending defense with any other user. Different provers could be assigned to different indexes. An order could be set up to dictate when a prover must submit their proof to update the account status in a coordinated manner. To avoid double-spending attacks, when an account wants to change indexes, it will have to inform the prover of the old index via a signature on its request. This way, they can be sure not to include transactions from an account that has changed indexes even if they only read the blockchain when they prove their batch of transactions. It is exactly like the tree structure without needing a zk-EVM rollup. More coordination must be realized to ensure the avoidance of the double-spending attack. The Merkle trees are just reorganized to emulate the behavior of several rollups on one.

To further explain the delivery example, the accounts would have several associated states: the funds they hold, potentially their associated index, their tasks and their status (if they are a client/package or a driver). Transactions can then be a fiduciary exchange, an index change, or a drop/move request. In the case of a move request, the prover then updates the rollup status by assigning this request to a driver. This update is always done by proving that the change of state follows the rules dictated at the deployment of the smart contract. In the case of a zk-EVM rollup, a fair exchange protocol can be imagined, but since proving the position in the physical world on the blockchain is difficult, it seems that a reputation-based system is the most suitable at the moment. A reputation system could then be set up for drivers and included in the rollup state. This reputation can be exclusive to the rollup or derivative via a reputation based on the blockchain data.

#### 5.3.2. Dynamic Networks

When deploying moving devices such as drones, ground robots or autonomous vehicles (which can be viewed as IoE entities), the connection to the whole group and the blockchain is not always possible at all times. The use of a rollup can then make sense because it allows a device not to be permanently connected to the blockchain but only when the prover wants to write data and update the status on it.

In such a context, it is possible to build a rollup where several provers are coordinated with indexes as described above. If the disconnection or separation into two (or more) groups can be anticipated, the users can be separated into groups before disconnection. As an example, let us consider a surveillance mission where a drone swarm must split into two groups to cover a larger area. We will assume that the drones are already all included in a rollup, ➀ in Figure 6. Before splitting, each user will announce which group it wants to belong to ➁ (this can also be enforced by rules established in advance in the rollup smart contract). Merkle trees are then reorganized to take the separation into account ➂. Each group is now independent ➃. Each drone can then keep track of the transactions that are issued in its group until the end of the mission and the old rollup status as a temporary distributed database. Thus, if the prover refuses to update the rollup status at the end of the mission, the drones have all the necessary data to produce the proof and can ensure themselves this update. If a drone wants to change groups and can anticipate it, a threshold signature of more than half of the drones in its group can attest that it has informed them and agree. They can present this signature to the other group who will proceed to the change. This mechanism must avoid double-spending in both groups. At the end, or for each group’s upload, the global Merkle root can be recalculated by the smart contract.

This architecture allows for the management of disconnections known in advance and not always depending on the blockchain. It brings the possibilities of managing distributed databases while conserving blockchain security. It is similar to a payment channel or state channel, but it is dynamic and secured by Zero-Knowledge proofs instead of relying on a trusted third party.

#### 5.3.3. Privacy Management

At present, data are a valuable asset that companies, organizations and institutions want to protect. Their motivations can be numerous but are generally related to the security, exploitation or monetization of these data. It is therefore important to protect them effectively by allowing private management of their contents. Several actors may want to sell them, and a blockchain is a practical tool that meets all these criteria. The blockchain allows for decentralized and immutable storage, making it safe. Cryptographic protocols included make it secure, and the choice of a permissioned or permissionless blockchain can guarantee the ownership of the data.

For example, a global organization can gather different actors by country with a reputation system and set up a voting system or a common database for them. A fair exchange protocol can complete the system, so that one country can sell information to others. Identity management can be significantly improved with Zero-Knowledge proofs. Privacy and management of attributes, access or data can be managed via validiums or volitions. This can be used by a company, a group linked to a particular application or a connected home, for example, to exchange and store confidential information while keeping the same security as on Layer 1.

Rollups merge the advantages of public and private blockchains without retaining the disadvantages. As previously discussed, a rollup is a tool whose rules can be configured before its creation, like a new private blockchain. However, it keeps the same security as the one present on the main blockchain (Layer 1). We suggest creating a permissioned rollup on a permissionless blockchain, requiring the agreement of an authority or group to integrate the rollup. Security is still based on the public blockchain, which is often more decentralized than private blockchains. The state of the rollup can also be encrypted data using an homomorphic encryption if the data must be included on the blockchain. Data can be managed more centrally or by servers that are authorized to do so in the manner of a validium to keep information private. Zero-Knowledge proofs can also be used to achieve anonymous transactions, as in Zcash [62], which uses SNARKs. Validium has the advantage of obfuscating the data from the blockchain point of view but still being able to prove that data were included in it. It is possible to disclose a batch of data using only one Merkle proof by reorganizing the Merkle tree beforehand.

## 6. Conclusions

The Internet of Everything aims to use the data produced by the sensors of connected IoT objects. To do this, companies need to store and process these data, which are large amounts of information. These resources can then be exploited to improve our daily lives by automating actions or by providing us with suggestions and information.

A blockchain is a distributed, replicated and immutable ledger used to store information and transactions, avoiding a single point of failure vulnerability via decentralization and providing strong security and cryptographic tools. The use of blockchains can be beneficial to many other sectors not limited to finance, especially for IoE whose connected objects often have low computing power or storage capacity. Blockchains can be seen as promising tools whose properties would secure IoE data while connecting them to an already existing economic and automated model, and restoring trust and ownership to users. However, the amount of IoE data to be processed may be too large for a blockchain at the moment, either because of the required memory space or the necessary throughput. In this context, this paper presented the main Layer 2 solutions existing and compared them. The Layer 2 solutions offer increased scalability to the blockchain, allowing us to overcome its limitations.

Rollups, and more particularly zk-rollups, are a recent blockchain Layer 2 solution allowing the recentralization of the creation of blocks on this Layer 2 (without trust needed for block creation) while maintaining decentralized verification done on the main blockchain (Layer 1). We argue that zk-rollups also allow the management of a state, such as a database, to update this state outside of the blockchain and to have its transitions verified by the blockchain. The state can be stored on the blockchain or outside it, and the transition rules are defined in advance. It also increases the number of transactions per second and reduces their cost drastically. It enables the creation of new rules, potentially different from those applied on the Layer 1 blockchain, according to the needs of each application. Moreover, applications whose transaction volume is too large for a classical public blockchain can also benefit from it to communicate in a secure way by storing an ephemeral history.

These numerous advantages greatly improve the scalability of blockchains and make it possible for systems with low computing power or storage space to build upon the security brought by the blockchain. We believe that zk-rollups remove the last obstacles blocking the adoption of blockchains by IoT, and more generally by IoE. By allowing the creation of sub-networks according to precise rules defined for each application and using a secure network at low cost, zk-rollups pave the way to many possible applications of blockchains in the IoE and many areas previously considered inaccessible to blockchains due to their constraints. Thus, this work ends with the introduction of several general and promising new use cases connecting IoE to blockchains. These use cases illustrate the use and possible benefits of zk-rollups, paving the way to many possible applications for blockchains on the IoE.

## Figures and Tables

**Figure 1 sensors-22-06493-f001:**
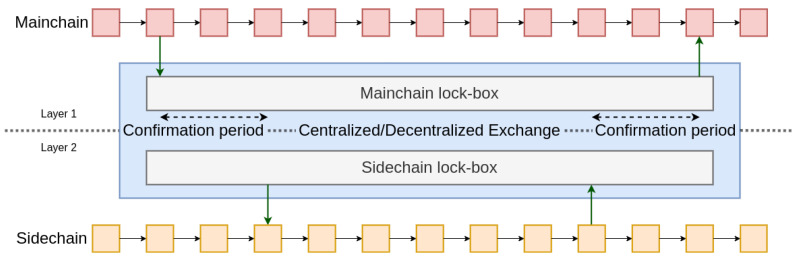
Illustration of a two-way-peg mechanism.

**Figure 2 sensors-22-06493-f002:**
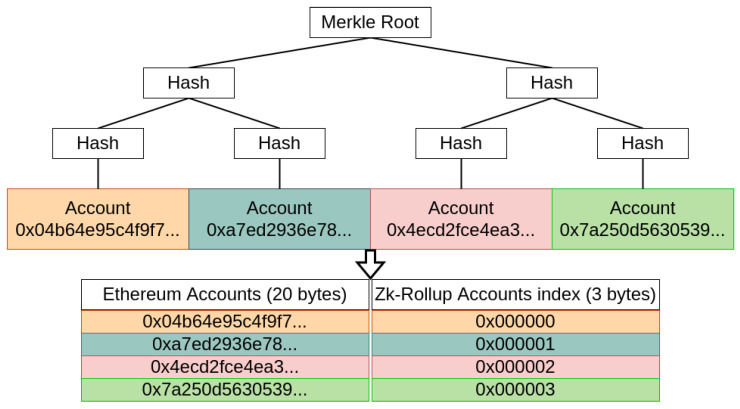
Representation of Ethereum accounts on a rollup.

**Figure 3 sensors-22-06493-f003:**
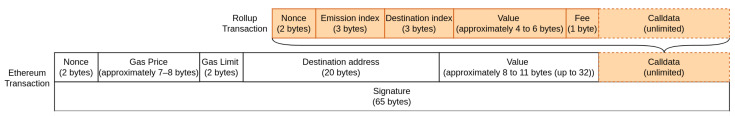
Integration of a rollup transaction into an Ethereum transaction.

**Figure 4 sensors-22-06493-f004:**
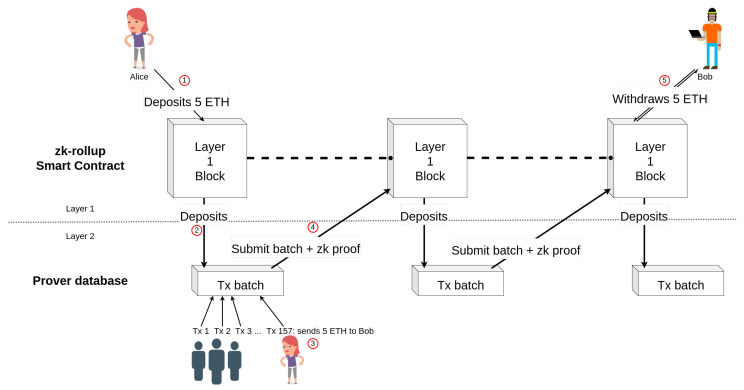
A transaction’s life cycle on a zk-rollup. Tx batch refers to a batch of transactions.

**Figure 5 sensors-22-06493-f005:**
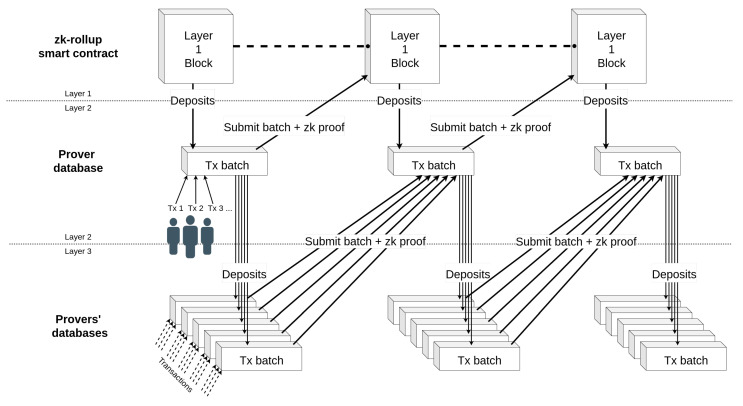
A tree structure of a zk-EVM zk-rollup.

**Figure 6 sensors-22-06493-f006:**
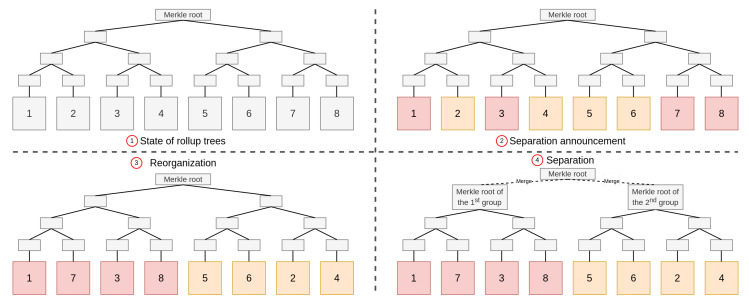
A separation on a rollup.

**Table 1 sensors-22-06493-t001:** Table inspired by [12,18] showing differences between blockchains and central databases.

Aspect	Permissionless Blockchain (Like Ethereum)	Permissioned Blockchain (Like Hyperledger Fabric)	Central Database (Like a Server)
Throughput (in TPS)	≈10–20	≈100–400	≈1000–5000
Latency (in seconds)	≈400–500	≈40–60	≈instantaneous
Number of readers	≈350,000–400,000	low but no theoretical limit	limited by server capacities
Number of writers	≈3000–10,000	≈10–20	1
Number of untrusted writers	High	Low	0
Centrally managed	No	No	Yes

## Data Availability

Not applicable.

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
