# Peer review of "Enabling Blockchain Services for IoE with Zk-Rollups"

_sensors, 2022, doi:10.3390/s22176493_

Round 1
Reviewer 1 Report
This manuscript provides an overview over blockchain layer 2 solutions, with a particular focus on Zk-Rollups. It is well written and the information contained therein is up-to-date.
Some applications to use Zk-Rollups in the context of Internet of Everything (IoE) are discussed. This is a very interesting theoretical idea, indeed. The current blockchain applications on large scale are limited due to the blockchain trilemma (decentralized, scalable and secure at the same time) and thus slow information processing, when compared to centralized solutions. Layer 2 or even layer 3 can be the answer to these problems.
I have the following issues to be addressed in revision so that the soundness of this study is potentially enhanced:
1. The discussion focusses mainly on Ethereum network and EVM compatible smart contracts. What impact the upcoming "merge" and switching Ethereum network on PoS could have?
2. What about other, faster, layer 1 solutions based on PoS: Solana, Avalanche and open source HyperLedger. Have other layer 2 solutions like Polygon and Loopring been considered?
3. When comparing Optimistic Rollups and Zk-Rollups, the Authors did not emphasize the fact that in the case of Zk-Rollups the existing smart contracts need to be significantly modified. This is not required in the case of Optimistic Rollups.
Reviewer 2 Report
- As for me the paper with no use of methods of research, presentation of research results and discussion is not an article, but rather a review paper; please consider the change of the status for the review paper;
- The whole paper is the literature review presented due to the subjective path of the Authors; please add the explanation of the methods used for this choice;
- In Chapter 5 the Authors present their opinion on the superiority of mathematical proofs in the paper, but with no arguments for; please add the concrete research results which allow for such a statement;
- The relation between customers, applications and intelligent solutions is not introduced in the paper; please explain them using the reference:
Kadłubek, M.; Thalassinos, E.; Domagała, J.; Grabowska, S.; Saniuk, S. Intelligent Transportation System Applications and Logistics Resources for Logistics Customer Service in Road Freight Transport Enterprises. Energies 2022, 15, 4668. https://doi.org/10.3390/en15134668
Reviewer 3 Report
1.Table 1 can be represented by data.
2.Please write the contribution of this research.
3.Please explain why you believe zk-rollups remove the last barrier to blockchain adoption for IoT, and more generally IoE.
4.In this context, this paper presented the main Layer 2 solutions existing and compared them. The Layer 2 solutions offer increased scalability to the blockchain allowing us to overcome its limitations. Rollups, and more particularly zk-rollups, are a recent blockchain Layer 2 solution al lowing the recentralization of the creation of blocks on this Layer 2 (without trust needed for block creation) while maintaining decentralized verification done on the main blockchain (Layer 1). Zk-rollups also allow the management of a state like a database, to update this state outside of the blockchain and to have its transitions verified by the blockchain. Please describe the author's contribution.
5.A rollup is a tool whose rules can be configured before its creation, like a new private blockchain. However, it keeps the same security as the one present on the main blockchain (Layer 1). Please explain why you suggest creating a permissioned rollup on a permissionless blockchain, requiring the agreement of an authority or group to integrate the rollup.
Round 2
Reviewer 2 Report
The authors improved the paper.
Author Response
The authors wish to thank the reviewer for its positive comment.
Reviewer 3 Report
The manuscript can be accepted for publication.
Author Response
The authors wish to thank the reviewer for its positive recommendation.